# Satellitome Analysis on *Talpa aquitania* Genome and Inferences about the satDNAs Evolution on Some Talpidae

**DOI:** 10.3390/genes14010117

**Published:** 2022-12-31

**Authors:** Juana Gutiérrez, Gaël Aleix-Mata, Eugenia E. Montiel, Diogo C. Cabral-de-Mello, Juan Alberto Marchal, Antonio Sánchez

**Affiliations:** 1Departamento de Biología Experimental, Área de Genética, Universidad de Jaén, Paraje de las Lagunillas s/n, 23071 Jaén, Spain; 2Departamento de Biologia Geral e Aplicada, Instituto de Biociências/IB, UNESP—Universidade Estadual Paulista, Rio Claro, São Paulo 13506-900, Brazil

**Keywords:** insectivora, satellite DNAs, satellitome, genus *Talpa*, Talpidae

## Abstract

In the genus *Talpa* a new species, named *Talpa aquitania*, has been recently described. Only cytogenetic data are available for the nuclear genome of this species. In this work, we characterize the satellitome of the *T. aquitania* genome that presents 16 different families, including telomeric sequences, and they represent 1.24% of the genome. The first satellite DNA family (TaquSat1-183) represents 0.558%, and six more abundant families, including TaquSat1-183, comprise 1.13%, while the remaining 11 sat-DNAs represent only 0.11%. The average A + T content of the SatDNA families was 50.43% and the median monomer length was 289.24 bp. The analysis of these SatDNAs indicated that they have different grades of clusterization, homogenization, and degeneration. Most of the satDNA families are present in the genomes of the other *Talpa* species analyzed, while in the genomes of other more distant species of Talpidae, only some of them are present, in accordance with the library hypothesis. Moreover, chromosomal localization by FISH revealed that some satDNAs are localized preferentially on centromeric and non-centromeric heterochromatin in *T. aquitania* and also in the sister species *T. occidentalis* karyotype. The differences observed between *T. aquitania* and the close relative *T. occidentalis* and *T. europaea* suggested that the satellitome is a very dynamic component of the genomes and that the satDNAs could be responsible for chromosomal differences between the species. Finally, in a broad context, these data contribute to the understanding of the evolution of satellitomes on mammals.

## 1. Introduction

Eukaryotic genomes are divided into two regions, one called euchromatin, where most genes are located [1] and heterochromatin, composed of a large number of repetitive DNA sequences. The repetitive sequences possess high variability in nucleotide base pair composition and number of copies, being involved in the genome size variation and diversity across three of life [2]. As a whole, the repeated DNA content (repeatome) includes dispersed, represented by all classes of transposable elements (TEs) and tandem satellite DNA (satDNA) repeats [3,4,5,6]. The satDNA sequences content of a genome, or satellitome [7], are constituted by non-coding repeated sequences organized in tandem, mainly located in constitutive heterochromatin regions, that in multiple species are observed on pericentromeric and centromeric regions of chromosomes and on a lesser extent on telomeres [2,8,9]. Satellite DNAs could represent a large proportion of the whole eukaryotic genomes, with variation between 0.1% and up to 50% (Revised in [2]). Due to the absence of general selective forces, the rate of evolution of the satDNA sequences is very high, and they show great variation in monomer size, nucleotide composition, chromosomal distribution, and abundance in the genome. Those factors can affect genome composition, structure, or size between species [10]. On many occasions, satDNA can be chromosome- or species-specific and can give rise to large genome variations between closely related species [2,11].

The biological significance of satDNA and heterochromatin itself is a topic that is currently arousing great scientific interest. Traditionally it has been considered as silenced junk DNA lacking any relevant function. However, there are multiple examples in insect, plant, or mammalian species of centromeric or telomeric repeat sequences being transcribed [8,10,12]. Transcription of satDNA could be associated with the heterochromatin assembly mechanism, the regulation of centromere function, the sister chromatid cohesion, chromosome organization, pairing and segregation, control of telomere elongation, gene regulation, karyotypic evolution, and speciation [4,5,13,14,15,16,17,18,19,20,21,22,23].

Massive sequencing techniques combined with computational and molecular cytogenetic analysis have opened a new way to characterize the repetitive DNA present in a genome. The development of powerful graph-based bioinformatics tools such as RepeatExplorer and TAndem REpeat ANalyzer (TAREAN) using massive sequencing data have enabled repeatome analysis [24,25,26]. These programs allowed the characterization of the repeated DNA content of the genomes from Next Generation Sequencing data. In this way, the satellitomes have been studied using these approaches in several animal [7,18,27,28,29,30,31,32] and plant species [33,34,35,36], allowing the testing of satDNA evolution models across distinct eukaryote groups. With the use of these new tools, it is possible to gain knowledge of the different families of satDNAs and to obtain information on the size and variability within a particular genome, as well as its evolution and genetic functions. In addition, this methodology allowed the detection of a large number of satDNA families at once. Illustrative examples are *Triatoma infestans*, *Locusta migratoria*, *Rhynchophorus ferrugineus*, and *Pontastacus leptodactylus* with 42, 62, 112, and 258 families identified, respectively [7,18,31,32].

The genus *Talpa* (family Talpidae) includes 11 species of fossorial mammals [37]. The species of the Talpidae family have very stable karyotypes [38,39]. Most of the species of *Talpa* have chromosome number 2n = 34 (exceptions *T. caeca*: 2n = 36 and *T. caucasica*: 2n = 38), and autosomal fundamental number (FNa) varies between 62 and 66 [39,40,41,42]. The sex chromosomes are also well conserved, with the X chromosome typically a medium-sized submetacentric or metacentric and the Y chromosome in most species is a dot-like metacentric, submetacentric, or acrocentric [42,43].

Analysis of repetitive DNAs identified heterochromatin on centromeres of most chromosomes of the karyotype of *T. romana* and *T. europaea* [42], while is less abundant in *T. aquitania* and *T. occidentalis* [37,40]. In addition, in some species, the Y chromosome is completely heterochromatic, and most of the species also present an autosomal pair with non-centromeric heterochromatin [37]. Although, the specific molecular content of the heterochromatin and chromosomal distribution of repeats is poorly known in the genus. Previous studies investigated only the chromosomal distribution of ribosomal genes, telomeric sequences, and a LINE fragment [37,42,43,44]. Telomeric sequences are located at chromosome ends but also arranged as ITSs (interstitial telomeric sequences) in some pericentromeric regions in *Talpa romana*, *Talpa europaea*, and *T. aquitania* [37,42]. LINE sequences are widely distributed on all chromosomes, as expected, with a noteworthy enrichment on pericentromeric C-band positive regions and the Y chromosome [43]. Interestingly, chromosome painting analyses demonstrate that heterochromatic blocks from two autosomal pairs and Y-heterochromatin share repeat sequences [43]. Recently, the genome of *Talpa occidentalis* has been sequenced, revealing that about 30% of the genome is comprised of repetitive sequences, similar data to other species in this group (15% of LINEs; 7% of SINEs; 4% of LTR; about 1% DNA transposon) [45]. 

In most of the mammalian genomes sequenced, little information is provided about the repetitive DNA sequence composition, the repeatome, and especially about the satDNA sequence composition, the satellitome. Until now, few satellitome analyses have been performed on mammal species [46], including insectivore species. Moreover, in this family, there is scarce information about repeat in general and satDNAs in particular. Data on satDNAs repeats remain almost completely missing for the *Talpa* species genomes. The aim of this work is to characterize the satDNA families, the satellitome, in the genome of the recently described species of this genus *T. aquitania* by using the Next Generation Sequencing approach. The satellitomes of the *T. aquitania* genome present 16 different satDNA families, including the telomeric sequences. Most of the satDNA families are present in the genomes of the other *Talpa* species analyzed, while in the genomes of other more distant species of the Talpidae family only some of them are present. Moreover, chromosomal localization by Fluorescence in situ Hybridization (FISH) revealed that some satDNAs are localized preferentially on centromeric and non-centromeric heterochromatin in *T. aquitania* and also in the sister species *T. occidentalis* karyotype. The differences observed between *T. aquitania* and the close relative *T. occidentalis* suggested that the satellitome is a very dynamic component of the genomes.

## 2. Materials and Methods

### 2.1. Talpa Specimens

For this study, we analyzed samples from three *T. aquitania* (captured in Torme (Burgos); northern Spain) and four *T. occidentalis* (captured in Granada; southern Spain) males. Permission for capture was granted by the Servicio Territorial de Medio Ambiente de Burgos (Junta de Castilla-León) and the Delegación Territorial de la Consejería de Agricultura, Pesca y Medio Ambiente de Granada (Junta de Andalucía). All capture and sacrifice protocols were approved by the Junta de Andalucía Ethics Committee for Animal Experimentation (code: 22/05/2018/094). 

### 2.2. DNA Extraction, Genome Sequencing, Sequence Clustering, and Analysis

Liver tissue samples were stored in 100% ethanol at −20 °C and DNA samples for PCR were extracted following the standard phenol-chloroform procedure. For Illumina sequencing, we extracted genomic DNA (gDNA) from one adult male of *T. aquitania* and one adult male of *T. occidentalis* using the Gentra Puregene Tissue Kit (Qiagen). The extracted gDNAs were sequenced with Illumina technology. Briefly, for genome sequencing, approximately 3 µg of genomic DNA was used for the construction of a library of 350-bp-length fragments. This library was based on the Illumina^®^ Hiseq™ 2500 platform and paired-end reads with 2 × 150 bp were obtained. A total of 4.5 Gb of sequences were obtained from the *T. aquitania* genome, corresponding to a coverage of about 2 × of the genome, considering the genome of similar size to the *T. occidentalis* genome (2.099 Gb) [45], and 2 Gb were obtained from the *T. occidentalis* genome (approximately a coverage about 1×). Graph-based clustering analysis [24,25,26,47], a method for similarity-based clustering of sequence reads, was performed with 829.778 reads of the *T. aquitania* genome using RepeatExplorer, implemented within the Galaxy environment (http://repeatexplorer.org/, accessed on 19 April 2021). 

The genome proportion for the repetitive DNA clusters was calculated as read percentages. Additionally, all the clusters were analyzed with sequence-similarity searches of the assembled contigs against GenBank using BlastN (http://www.ncbi.nlm.nih.gov/, accessed on 5 September 2021) and Repbase using the program RepeatMasker version 4.0.9 (http://repeatmasker.org; https://www.dfam.org/, accessed on 1 April 2022, [48] and CENSOR (http://www.girinst.org/, accessed on 1 April 2022). In addition, to identify satDNA repeats, contigs were analyzed using Dotmatcher (http://emboss.bioinformatics.nl/cgi-bin/emboss/dotmatcher/, accessed on 5 September 2021). For the analysis of the characteristics of sequences of the clusters, we used the software Bioedit (version 7.2.5) (http://www.mbio.ncsu.edu/BioEdit/bioedit.html, accessed on 5 September 2021) [49] and Clustal Omega (https://www.ebi.ac.uk/Tools/msa/clustalo/, accessed on 5 September 2021) [50]. 

Estimates of evolutionary divergence between nucleotide sequences were conducted with MEGA 10 using *p*-distance. The satDNA family abundance and divergence were calculated with RepeatMasker with the “-a” option and the RMBlast search engine. Four million reads of *T. aquitania* and *T. occidentalis* samples were selected and aligned to the total collection of satDNA dimers, or monomer concatenations of approximately 200 bp in length in satDNAs with small size monomer. A satellite landscape was generated considering distances from the sequences applying the Kimura 2-parameter model with the Perl script calcDivergenceFromAlign.pl and createRepeatLandscape.pl from the RepeatMasker suite. The tandem structure index (TSI) was calculated from the out file of RepeatMasker output as in Montiel et al. [31]. We calculated the number of reads with at least 89% read length aligned to the dimer (internal/pure satDNA reads) divided by the total number of aligned reads (internal/pure + external/mixed satDNA reads). The satellite landscape created was used to calculate the divergence peak (DivPeak) and the relative abundance size of the peak (RSP) for each satDNA family following the method in [51]. Briefly, the divergence value where the landscape had a maximum (DivPeak) was calculated and the relative peak size as the sum of satDNA abundances at ±2% divergence from the divergence peak (peak size) was divided by the total abundance of that satDNA family. The relative abundance among samples was calculated as the log_2_ of the ratio of the compared sample divided by the *T. aquitania* sample in order to facilitate the understanding of the fold-change between samples. Then, to check the presence of *T. aquitania* satDNAs on the closest relative species the consensus sequences were also masked on the chromosome-scale genome assembly of *T. occidentalis*, based on long- (PacBio) and short-read (Illumina) sequencing and scaffolded using Hi-C data, published by Real et al. [45] (GCA_014898055.1). Moreover, aiming to check sharing of satDNAs among representatives of family Talpidae abundance and divergence were calculated from low-coverage sequencing data of other species that were available on NCBI SRA database: *Talpa europaea* (SRX8240408), *Scalopus aquaticus* (SRX4562103), *Condylura cristata* (SRX101046), *Galemys pyrenaicus* (SRX10243621) and *Uropsilus gracilis* (SRX4562112). Calculations and statistical analysis as a correlation between variables using Spearman’s rank correlation rho and comparison between paired samples using Wilcoxon signed rank exact test were performed in R base v.4.0.1 [52]. Figures were also plotted in R with ggplot2 and Viridis packages [53,54].

### 2.3. Satellite DNA Probes, Chromosome Preparations, and FISH

The consensus sequence of each satDNA family (Appendix A) was used to design one (small consensus sequences) or two oligonucleotides (large consensus sequences) by using the Primer3 software [55] (Appendix A). Only three SatDNA families with large monomers (TaquSat4-437-466; TaquSat5-3102 and TaquSat11-71) were amplified and labeled by PCR using the specifically designed primer pairs. PCR was performed in 50 µL of reaction mixture containing 100 ng of *T. aquitania* genomic DNA, 10 pmol of each primer, 1 µL 10 mM dNTPs, 5 µL of 10× NH_4_ Reaction Buffer, 2.5 µL of 50 mM MgCl_2_, and 5 U of BIOTAQ™ DNA Polymerase (Bioline). The PCR program used was 5 min at 95 °C and 30 cycles: 30 s at 95 °C, 30 s at 55 °C, and 60 s at 72 °C, with a final extension of 5 min at 72 °C. PCR amplicons were resolved in ethidium bromide-stained 1% agarose gels; the appropriate bands were isolated from the gel with a QIAquick gel extraction kit (Qiagen), and labeled with biotin-16-dUTP or digoxigenin-11-dUTP by PCR amplification, with the specific primers pairs. For satDNA families with small monomers, the oligonucleotides based on the most conserved regions were directly labeled with biotin-16-dUTP or digoxigenin-11-dUTP using Terminal Transferase (Roche) as described by Pita et al. [18]. 

Chromosomes were prepared from bone marrow cells following the method described by Burgos et al. [56]. The location of satDNAs sequences by Fluorescence in situ Hybridization (FISH) was performed as previously described by Pita et al. [18], Fernández et al. [57], and Cabral-de-Mello and Marec [58]. Briefly, the hybridized probes, labeled with biotin-16-dUTP, were detected by avidin-based indirect fluorescence techniques, with two rounds of amplification [18,57] or with Alexa Fluor 488 streptavidin conjugate, while the probes labeled with digoxigenin-11-dUTP were detected with Anti-Digoxigenin-Rhodamine (Roche) [58].

Slides were mounted with Vectashield (Vector Laboratories, Newark, CA, USA) containing DAPI to counterstain the chromosomes. The images were captured and analyzed using a fluorescence microscope (Olympus BX51) equipped with a CCD camera (Olympus DP70), and processed with Adobe Photoshop software.

## 3. Results and Discussion

### 3.1. General Characterization of T. aquitania Satellitome

Sequencing data (4.5 Gb) of the *T. aquitania* genome produced 30,149,458 reads. Nucleotide analyses showed that A + T genome content was 59.06% (GC content 40.94%). A high A + T content is a common feature within the Talpidae genomes (58.09% *T. occidentalis*, 59.5% *T. europaea*, 58.3% *G. pyrenaicus,* 58% *S. aquaticus,* 58.1% *C. cristata*, and 58.9% *U. gracilis*) (GenBanK data; [45,59]).

From the total reads, a subset of 8 million pair-end reads was randomly selected and processed in RepeatExplorer. From that subset, 829,778 sequences were selected in the pipeline. Those correspond to about 6% of the genome of *T. aquitania*, considering the genome of similar size as the *T. occidentalis* genome (2.099 Gb) [45]. 

After RepeatExplorer clustering, 263,881 reads were grouped into 38,295 clusters, with 20% of the *T. aquitania* genome composed of repeated sequences. All the remaining reads (565,897) were classified as singletons. From those clusters, 425 represented at least 0.0024% of the genome and those that featured a star-like or circular graph topology, typically observed in the satDNA families, were deeply analyzed. For each candidate cluster, we examined the contigs assembled by RepeatExplorer to search tandem repeated structures using the Dotmatcher and multiple-sequence alignments and manual inspection to determine the consensus sequences. After the computational analysis, we identified 15 satDNA families (Table 1 and Appendix A). The satDNA families were named according to Ruiz-Ruano et al. [7], including the species name abbreviation, a number in decreasing order of abundance and the length of the repeat sequence monomer, starting from TaquSat1-183 (the most abundant) to TaquSat15-64 (the least abundant). For the TaquSat4 family, two variants were found with 437 and 466 bp. Telomeric sequence repeat TTAGGG was not found through this analysis. Nevertheless, the (TTAGGG)_50_ repeat was included in the RepeatMasker analysis showing that the abundance of telomeric repeats was 0.13% (Table 1). In this way, including the telomeric sequences, the *T. aquitania* satellitome is composed of at least 16 satDNA families, corresponding to 1.24% of the genome (Table 1). This percentage of satDNA is in accordance with observed in other Talpidae species as *G. pyrenaicus* (1.08%) [59]. On other species of mammals in which RepeatExplorer analysis was carried out, Valeri et al. [46] found only one satDNA family, corresponding to 0.87% of the genome of *Trichechus manatus*, revealing that the picture of abundance and number of families could be highly divergent on distinct groups, deserving more analysis.

The first satDNA family TaquSat1-183 represents 0.558% of the genome. Together with the next five families in the list and the telomeric sequences, they all comprise 1.13% of the genome, while the remaining 11 satDNAs represent only 0.11%. The mean A + T content is 50.43% (with variation between 67.2% and 30.4%), lower than the overall genome A + T content (59.06%). The mean of the monomer length is 289.24, with variation between the largest one with 3102 bp (TaquSat5-3102) and the smallest one with 6 bp (TaquSat3-6 and telomeric sequences), being the monomer length of the most abundant satellite DNA of 183 bp (TaquSat1-183). Most of the satDNA families have repeat units smaller than 183 bp, with the exception of the TaquSat4-437-466 and TaquSat5-3102 in global the mean of monomer length is 289.24 bp (median 71.00).

The relationships between satDNA families were analyzed by comparison of the consensus sequences. Most of the satDNA families did not show similarity with the sequences of other families. However, in the satDNA TaquSat4-437-466, analyzing the contigs alignments, and manually, we identified two different monomer lengths 437 and 466, both monomers shared a fragment with an identity of 78.21%, and the remaining portion of the longer monomer corresponded with the first portion of a new the monomer repetition. On some posterior analysis, they are considered as TaquSat4-437 and TaquSat4-466.

Nucleotide divergence of satDNA families in *T. aquitania* ranged between 0.14% (TaquSat5-3102) and 21.69% (TaquSat14-17). Altogether, the satellitome of *T. aquitania* shows a mean nucleotide divergence value of 13.07% (Table 1). Additionally, we calculated the tandem structure index (TSI) of each satDNA [31]. This index represents the proportion of internal reads of satDNA arrays (reads containing only satDNA sequence) with respect to the total of reads with satDNA sequence. The TSI varies between 0 and 1 and informs about the clusterization level of the satDNA family. Therefore, dispersed families with arrays formed by a low number of repeat units would present a lower value of TSI in comparison to families with their repeat units concentrated in longer arrays, which would present TSI values closer to 1. The satDNAs of *T. aquitania* have a TSI mean of 0.53. The first six most abundant satDNAs (including the telomeric) have the highest TSI values, ranging between 0.99 for TaquSat5-3102 and 0.76 for TaquSat3-6. However, in satDNA families with large monomer sizes, high values of the TSI index could be related more to the size than the structure since short reads used in that calculation could correspond to the same monomers and not to tandem monomers. The remaining ones have the lowest TSI values ranging between 0.53 for TaquSat6-84 and 0.00 for TaquSat15-64.

The satellite landscape is a good representation of both the homogenization or degeneration status of each satDNA family. Those landscapes were obtained by plotting the abundance versus divergence of satDNA sequences against their consensus sequence. SatDNA families with a recent expansion would be formed by nearly identical monomers and almost the total abundance of the family would be contained in the landscape peak. These families would show leptokurtic satellite landscape distributions, with their peaks on low divergence values, as the families TaquSat5-3102, TaquSat8-45, and TaquSat11-71 (Figure 1). On the contrary, satDNA families with high divergence would contain monomers with a great accumulation of mutations in their sequence and the satDNA family abundance would not be contained in the peak. These divergent families would show platykurtic satellite landscape distribution with the peak on high divergence values. Platykurtic landscapes were shown by the family TaquSat3-6, TaquSat7-60, and TaquSat13-54 (Figure 1). The rest of the satDNA families displayed mesokurtic satellite landscapes indicating families amplified time ago and in process of degeneration (Figure 1). In addition, we calculated two indices, the divergence value where the landscape had a maximum (DivPeak) (Table 1), as a measure of the family degeneration, and the relative abundance size of the peak (RSP) (Table 1), as a measure of the family homogenization [51]. Both indices were inversely correlated in the *T. aquitania* satellitome (Spearman’s rank correlation r_s_ = −0.75, *p* < 0.001) where the recently expanded families showed low DivPeak and high RSP values, and the highly divergent and degenerated families presented high DivPeak and low RSP values (Figure 2).

### 3.2. Sister Species Genomic Analysis Reveals Insight about satDNAs Diversification on Talpidae

The satellitome is a very dynamic genome component that could vary between closely related species or even among populations of the same species [15,18,31]. We were interested in studying the putative conservation of satDNAs in *T. occidentalis,* the species considered as the closely related species of *T. aquitania* [37,60,61] and other Talpidae representatives. Therefore, the consensus sequences of *T. aquitania* satDNA families were used to investigate their presence in the species *T. occidentalis* and other Talpidae.

The abundance and divergence of each satDNA family sequence were analyzed with RepeatMasker using a sample of 4 million reads of *T. occidentalis* (aprox. 400 Mb, 0.2× genome coverage). Fifteen of the sixteen satDNAs found on *T. aquitania* genome were shared with *T. occidentalis*, with the unique exception of TaquSat8-45. The satDNA sequences accounted for 0.69% of the *T. occidentalis* genome, corresponding to almost half of the satellitome in *T. aquitania*. The variation of abundance among samples was calculated with the ratio between samples (*T. occidentalis*/*T. aquitania*), which was represented as log_2_(*T. occidentalis*/*T. aquitania*). Most satDNA families (10 out of 15) were less abundant on the *T. occidentalis* genome (log_2_ ratio < −0.6). The most abundant family of *T. aquitania*, TaquSat1-183, only accounted for 0.13% of *T. occidentalis*, being the third satDNA family in terms of abundance on *T. occidentalis*. In contrast, the two variants of TaquSat4 (TaquSat4-437 and TaquSat4-466) were more abundant on *T. occidentalis* (log_2_ ratio > 0.6), with TaquSat4-437 being the most abundant satDNA family on *T. occidentalis* accounting for approximately the 0.20% of the genome (Appendix A). As fast-evolving components of the genome, the changes in satDNA abundance among closely related species are not uncommon. The expansion or divergence of these repeated sequences configures specie-specific genetic backgrounds, which could contribute to the speciation process [15,62].

The satellite landscapes for each family were also compared between species, including the degeneration (DivPeak) and homogenization (RSP) indices. The landscape distribution of satDNA families did not show significant differences, besides abundance, between species, and also indices, DivPeak and RSP, were similar among samples (Table 1 and Appendix A). It was interesting, however, that satDNA families with a leptokurtic distribution on their satellite landscape on *T. aquitania*, recently expanded families, were low abundance with larger degeneration value (TaquSat11-71) and low (TaquSat5-3102) or even not present (TaquSat8-45) on the homogenization index on *T. occidentalis*. In fact, one would expect that between closely related species, the satellitome would be similar, and between species that are more distant, the satellitome might show significant differences. However, in a comprehensive analysis of the satellitomes of bird species, the observed pattern was not in agreement with this assumption. In a group of birds with more deeply diverged species, satellites were found to be more similar and were less similar among more recently diverged species [63]. 

Additionally, as the first genome assembly of *T. occidentalis* was published by Real et al. [45], satDNA consensus was mapped to the assembly to test their presence on it. The proportion of satDNA sequences in the assembly was 0.20%, a lower value than the estimation with *T. occidentalis* low-coverage sequencing data (0.69%). Studies identifying satDNA sequences on the assembled genomes are scarce but their underestimation of the assemblies has been a common theme. For instance, the satellitome of *Rhodnius prolixus*, which accounts for 8% of the genome, only makes up 5.6% of the genome assembly [30]. Another example is the satellitome of the red palm weevil, *R. ferrugineus*, which is 10 times underrepresented in the genome assemblies [31]. The abundance variation of each family among *T. occidentalis* sequencing data and the assembly was calculated as above (log_2_(*T. occidentalis* assembled genome/*T. occidentalis* genome)) (Appendix A). Even though no significant differences were found between the two samples (*p* = 0.1167 two-sided Wilcoxon signed-rank test), there was a negative correlation between the TSI value and the variation of abundance among samples (Spearman’s rank correlation r_S_ = −0.63 *p* = 0.008). Thereupon, satDNA families with larger TSI values, families with higher clusterization forming large arrays, were less abundant in the assembly, while satDNA families with lower TSI values, dispersed families with arrays made by a low number of monomers, were more abundant (Appendix A). Repeated DNA sequences tend to be discarded in the assembly process, due to the challenge that their assembly supposes. Therefore, satDNA families dispersed in small clusters scattered in the genome were better assembled than satDNA families composed of large clusters that were likely to be collapsed just in a few monomers. The correlation between TSI and the satDNA estimation in genome assemblies has been reported before. In *R. ferrugineus*, the high-TSI families showed a higher underestimation than low-TIS families, for example, the family RferSat01-169, which accounts for 0.01–0.06% of the assemblies whereas it was estimated at 20% of the genome [31].

The presence of *T. aquitania* satDNA families was tested on the genomes of other species of the family Talpidae, including *T. europaea*, *S. aquaticus*, *C. cristata*, *G. pyrenaicus*, and *U. gracilis*. Raw data of these species were available in the NCBI SRA database and the analysis was performed as above. The number of satDNA families that present those species was variable. On *T. europaea*, the other member of the subfamily Talpinae, it was identified 14 satDNA families, one less than in *T. occidentalis* (Appendix A). The species where fewer families were identified were *U. gracilis* (7 families), the most distant species from *T. aquitania*. Interestingly, those seven families are also presented in the other species and may be conserved in the family Talpidae. The abundance and divergence of satDNA families also seem to be related to the phylogenetic distance to *T. aquitania*. Therefore, for *T. aquitania* satDNA sequences, *T.* europaea displayed a larger abundance (2.27%) and lower Kimura divergence (15.65%), while *U. gracilis* showed low abundance (0.03%) and larger Kimura divergence (27.21%). Analysis of Dugongidae mammals revealed also the conservation of one satDNA family found initially in *T. manatus* in other species of the family with differences in abundance [46]. Those differences would be consistent with the library hypothesis proposed by Fry and Salser [64]. From the same pool of satDNA sequences, certain families could eventually expand and fix in one species, whereas other families could expand in other species. Consequently, at the same time species distance themselves, their satellitomes also do so, contributing to differences among species or even favoring new speciation events [8,65]. 

### 3.3. SatDNAs Chromosome Localization on T. aquitania and T. occidentalis Reveals Information about Karyotypic Evolution

*T. aquitania* karyotype has a diploid number of 2n = 34 (NFa = 64) and all chromosomes including the sex chromosomes are biarmed, either metacentric or submetacentric. The X chromosome is medium-sized submetacentric and the Y chromosome is submetacentric and the smallest of the karyotype. As in all the karyotypes of all *Talpa* species studies on *T. aquitania*, one of the largest autosomes (pair 3) harbors a secondary constriction [35,38,40]. In *T. aquitania* karyotype C-banding revealed heterochromatin in centromeric regions of a few pairs of chromosomes, on the proximal part to the centromere of the short arm of pair 14, on the entire short arm of pair 16, and in the entire Y chromosome [35]. The *T. occidentalis* karyotype is very similar to the *T. aquitania,* with the main differences due to the size of the Y chromosome, which is a medium size biarmed submetacentric on *T. aquitania* and a dot-like probably metacentric, in *T. occidentalis*; and in the size of the small arm of the autosome pair 16, that is large on *T. aquitania* and very small in *T. occidentalis* [35,41]. 

Only seven of the satDNA families of *T. aquitania* were visualized by FISH in the chromosomes of this species and only six on *T. occidentalis* chromosomes. These satDNAs presented very similar distribution on both species, and most of the variations are due to the above-commented differences between the karyotypes (Figure 3).

Three satDNAs (TaquSat1-183, TaquSat3-6, and TaquSat4-437-466) are located on pericentromeric regions, including the X chromosome, however, only TaquSat4-437-466 is also located on the Y chromosome, and TaquSat3-6 has one interstitial band in the arm of one small autosome in *T. aquitania* not present in *T. occidentalis* (Figure 3a–f).

SatDNA TaquSat2-107 is located on subtelomeric regions of the long arm of the Y chromosome and on the subtelomeric regions of the heterochromatic short arm or the pair 16 in *T. aquitania* (Figure 3g), while in *T. occidentalis* it is probably on the small arm of the minute Y chromosome and occupying all the small arm of the pair 16 (Figure 3h). These localizations suggested that the long arm of the Y chromosome of *T. aquitania* arises by the enlargement of the small arm of the Y chromosomes of *T. occidentalis.* In the same way, the enlargement of the heterochromatic small arm of a pair 16 of *T. occidentalis* arises the big small arm of this pair in *T. aquitania.* Heterochromatin enlargements in both chromosomes amplified the satDNA TaquSat6-84. In fact, TaquSat6-84 is accumulated on the heterochromatin of both, the Y chromosome and the short arm of the pair 16 in *T. aquitania*, but was undetectable in *T. occidentalis* (Figure 3i,j). The recent amplification of pair 16 heterochromatic small arm in *T. aquitania* could explain the different composition with chromosome 14 heterochromatin.

SatDNA TaquSat5-3102 is distributed along all the chromosomes in both species, with enrichment on heterochromatin regions of pair 14, the centromeric region of some pairs, and on the Y chromosome; however, is not accumulated on the heterochromatin region of pair 16 (Figure 3k,l). The satDNA TaquSat11-71 is also distributed in both species along all the chromosomes, with enrichment on heterochromatin short arms of pairs 14 and 16, and on the Y chromosome (Figure 3m,n). Hence, the heterochromatic short arm of pairs 14 and 16 on both species shared satellite sequences, as the TaquSat2-107 and TaquSat5-3102, but also present different compositions on sequences, such as TaquSat6-84 and TaquSat5-3102.

Our data indicate that in *Talpa* genomes satDNA families are preferentially accumulated in the centromeric regions and in the C-band positive heterochromatic regions. Overall, the results demonstrate that the satellitome of *T. occidentalis* and *T. europaea* is very similar to that of *T. aquitania*, but differed significantly in the presence and amount of several satDNA families. As the estimated divergence time between *T. aquitania* and *T. occidentalis* is 2.47 ± 0.12 million of years ago (Mya) and between these species and *T. europaea* is 2.82 ± 0.10 Mya [61], the results demonstrated that the satellitome is a very dynamic component of the genomes due to the differences observed in these closely related species. 

## Figures and Tables

**Figure 1 genes-14-00117-f001:**
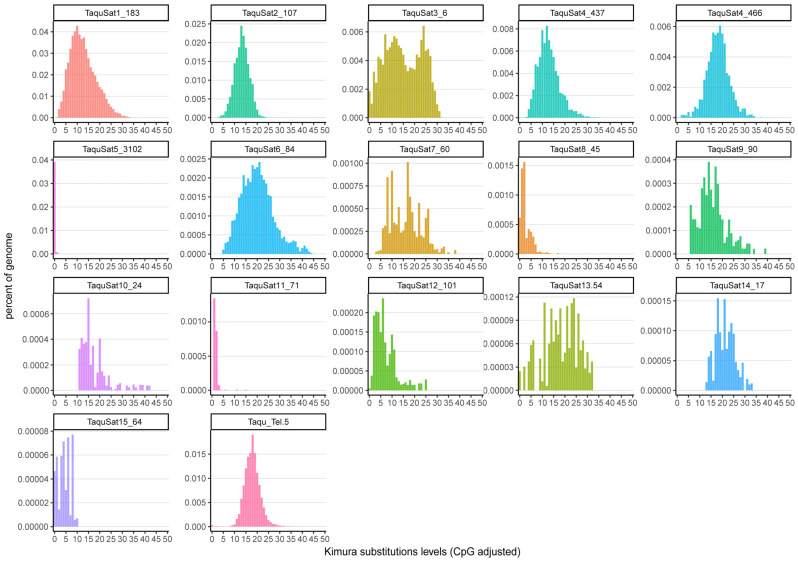
Satellite landscapes (abundance versus divergence) of *T. aquitania* satDNA families.

**Figure 2 genes-14-00117-f002:**
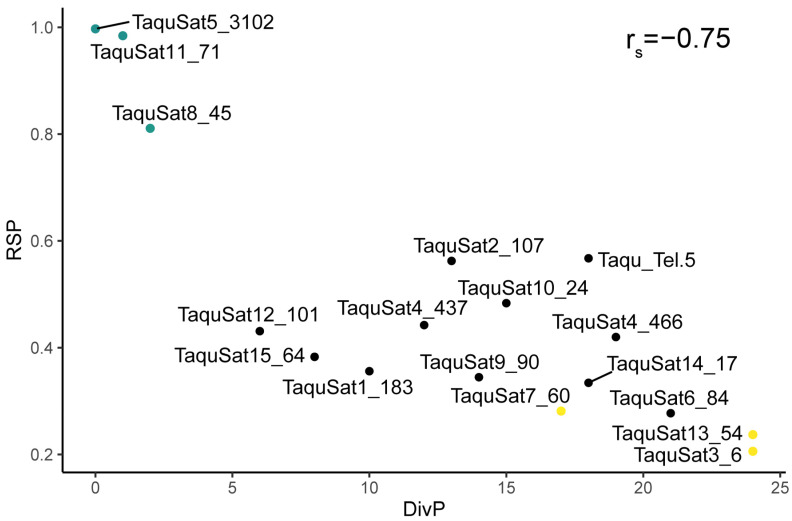
Correlation between the homogenization (RSP) and degeneration (DivP) indices of *T. aquitania* satDNA families. Recently expanded families are in blue and highly divergent and degenerated families are in yellow.

**Figure 3 genes-14-00117-f003:**
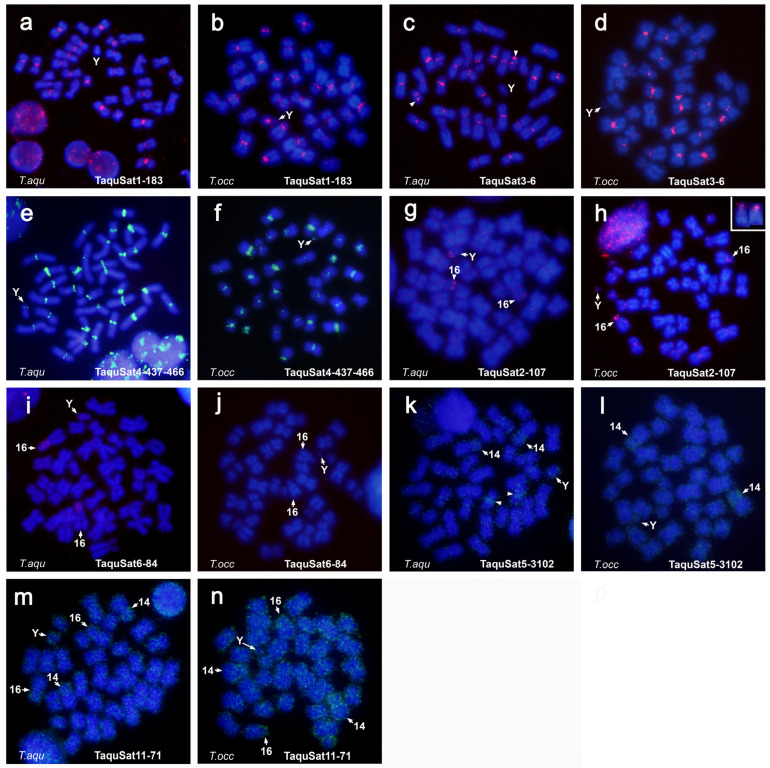
Male metaphases of *T. aquitania* (**a**,**c**,**e**,**g**,**i**,**k**,**m**) and *T. occidentalis* (**b**,**d**,**f**,**h**,**j**,**l**,**n**) hybridized with satDNA probes: (**a**,**b**): TaquSat1-183; (**c**,**d**): TaquSat3-6; (**e**,**f**): TaquSat4-437-466; (**g**,**h**): TaquSat2-107; (**i**,**j**): TaquSat6-84; (**k**,**l**): TaquSat5-3102; (**m**,**n**): TaquSat11-71. Arrowhead denotes the interstitial signal on the small autosome pair in (**c**), accumulation of signal on the centromeric region of one autosome pair in (**k**). Insert in (**h**), pair 16 chromosomes from other metaphase hybridized with the same probe.

**Table 1 genes-14-00117-t001:** Data of the satDNA families found in *T. aquitania*: genome proportion (%), the repeat unit length, A + T content and divergence (%), tandem structure index (TSI), divergence peak (DivP) and relative abundance size of the peak (RSP).

Name	Genome Proportion	Repeat Unit Length (bp)	A + T Percentage	Kimura Divergence	TSI	DivP	RSP
TaquSat1-183	0.55864	183	67.2	12.85	0.85	10	0.36
TaquSat2-107	0.16776	107	55.1	13.63	0.93	13	0.56
TaquSat3-6	0.12463	6	66.7	15.61	0.76	24	0.21
TaquSat4-437-466	0.146 (0.08–0.066)	437–466	30.4–32.6	13.3–18.5	0.83–0.84	12–19	0.44–0.42
TaquSat5-3102	0.04003	3102	53	0.14	0.99	0	1.00
TaquSat6-84	0.04000	84	65.5	20.63	0.53	21	0.28
TaquSat7-60	0.00993	60	48.3	16.72	0.41	17	0.28
TaquSat8-45	0.00532	45	51.1	3.01	0.45	2	0.81
TaquSat9-90	0.00425	90	56.7	16.28	0.40	14	0.34
TaquSat10-24	0.00418	24	50	18.16	0.48	15	0.48
TaquSat11-71	0.00233	71	40.3	2.1	0.09	1	0.98
TaquSat12-101	0.00180	101	50.5	7.73	0.01	6	0.43
TaquSat13-54	0.00175	54	59.3	18.7	0.20	24	0.24
TaquSat14-17	0.00129	17	35.3	21.69	0.36	18	0.33
TaquSat15-64	0.00045	64	45.3	4.89	0.00	8	0.38
Telomeric	0.13160	6	50	18.29	0.91	18	0.57
Total	1.2422						
Mean		289.24	50.43	13.07	0.53	13.06	0.48
SD		737.28	11.19	6.89	0.33	7.63	0.24
Median		71.00	50.50	15.61	0.48	14.00	0.42

## Data Availability

The datasets generated and analyzed during the current study are available on request from the corresponding author.

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
