# Peer review of "Satellitome Analysis on Talpa aquitania Genome and Inferences about the satDNAs Evolution on Some Talpidae"

_genes, 2022, doi:10.3390/genes14010117_

Round 1

Reviewer 1 Report

Gutiérrez et al. performed a satellitome analysis on Talpa and also analyzed the isolated satDNAs in other Talpidae species.

Results are ok and the images are nice. Overall, satellite DNA knowledge is growing in the last decade, but several groups remain little explored, like mammals. So, considering that authors performed a characterization and also a comparative study, I recommend this manuscript, but I have some minor comments and suggestions for authors

Abstract:

-       Line 12: “only cytogenetic data are actually available for the genome of this species”. I can see that the same research group published the mitogenome of this species Aleix-Mata et al. (2020, Mol Biol Rep). So, this statement is not necessarily true. Please, rephrase it.

-       Lines 14-16: percentage values show some issues here. Summing 0.558%, 1.13% and 0.11%, it exceeds the 1.24% of the genome. Please, confirm this.

-       Line 14: SatDNA is used here, but I think it should be presented as satellite DNA first

-       Line 17: better use median monomer length

Introduction

-       Line 43: Include the word “general” before “absence of selective forces”. There are several examples providing evidences that some satDNAs exhibit function, which could imply in selective forces. Indeed, these evidences could be cited here.

-       Line 52: “repeat sequences being transcribed refuted this paradigm”. Authors should take care with such assumption. Transcription does not reflect function. Sometimes, if an array of satDNA is near the 3’ end of a gene, it could be transcribed.

Materials and Methods

-       Line 115: replace “used” by “analyzed”

-       Line 131: Describe better the obtained data. “9Gb of sequences were obtained”. From both species? From each species?

-       Line 132: Coverage of 4x was obtained. I guess that this information is based on a genome of 1Gb of size? Am I correct? If so, where can we get this information? Please, cite

-       Line 166: Was this genome assemble with long reads (CLR, HiFi, ultralong ONT)? This information is important, especially because you discuss the underrepresentation of satDNAs in the genome assembly

-       Line 172: Which statistical analyses? R is just a software, you need to detail this information

Results and Discussion

-       Line 205: Something is wrong here. Multiplying 30M reads by 150bp, generate 4.5Gb.

-       Line 218: What is "deeply analyzed"? Please, provide a more detailed information, especially because this is the most important step of the pipeline (i.e. determining the monomer sequences of satDNAs)

-       Line 222: Names are different TaquSat or TaquiSat? Please, adjust this and check throughout the text

-       Line 223: correct “tow” by “two”

-       Line 236: correct “thee”

-       Lines 248-251: “we identified two different monomer length 437 and 466, both monomerS shared a fragment with an identity of 78.21%, and the remaining portion of the longer monomer corresponded with the first portion of a new the monomer repetition.” How this was done? On the contigs of RepeatExplorer? Which software? Manually? That's why authors should describe better the performed analyses to recover monomers of satDNAs

-       Line 263: I think we should interpret this specific result with care. Considering that monomer size of TaquSat5 is 3102bp-long, it is expected that TSI is the highest, mainly because fragments of Illumina sequencing are usually 300bp-long. So, the 0.99 value of TSI is most likely due to mapping in the same monomer, not mapping in the same satDNA adjacent monomer. Authors should consider this and highlight this on the text

-       Line 295: correct “populations”

-       Line 301: Did you take into account the divergence values? When stating that TaquSat8 represents 0.69% of T. occidentalis genome, are all the divergence values included? If so, this should be indicated on the methods section. From what I remember, RepeatMasker outputs divergence values until 70%.  

-       Line 380: if all chromosomes are biarmed, NF would be 68, right?

Author Response

Reply to the minor comments and suggestions for authors of the Reviewer 1

Abstract:

  • Line 12: “only cytogenetic data are actually available for the genome of this species”. I can see that the same research group published the mitogenome of this species Aleix-Mata et al. (2020, Mol Biol Rep). So, this statement is not necessarily true. Please, rephrase it.

We have included the term nuclear and the sentence now is: 

Only cytogenetic data are actually available for the nuclear genome of this species.

  • Lines 14-16: percentage values show some issues here. Summing 0.558%, 1.13% and 0.11%, it exceeds the 1.24% of the genome. Please, confirm this.

In fact, the sum did not correspond because the percentage of 1.13% includes the first family in abundance. Hence, the sum is 1.13 + 0.11= 1.24. We have made some changes on the sentence, and we hope that now could be clearer. The sentence now is: 

The first satellite DNA family (TaquSat1-183) represents the 0.558%. The six more abundant families, including TaquSat1-183, comprise 1.13%, while the remaining 11 sat-DNAs represent only 0.11%.

  • Line 14: SatDNA is used here, but I think it should be presented as satellite DNA first

OK, done

  • Line 17: better use median monomer length

OK, done

Introduction

 Line 43: Include the word “general” before “absence of selective forces”. There are several examples providing evidences that some satDNAs exhibit function, which could imply in selective forces. Indeed, these evidences could be cited here.

Ok, the word “general” is now included in the sentence, and references about the possible function of satDNA are included in the following paragraph.

  • Line 52: “repeat sequences being transcribed refuted this paradigm”. Authors should take care with such assumption. Transcription does not reflect function. Sometimes, if an array of satDNA is near the 3’ end of a gene, it could be transcribed.

OK, we have eliminated in final part of the sentence, now is:

However, there are multiple examples in insect, plant or mammalian species of centromeric or telomeric repeat sequences being transcribed [8,10,12].

Materials and Methods

 Line 115: replace “used” by “analyzed”

OK, done

-      Line 131: Describe better the obtained data. “9Gb of sequences were obtained”. From both species? From each species?

-      Line 132: Coverage of 4x was obtained. I guess that this information is based on a genome of 1Gb of size? Am I correct? If so, where can we get this information? Please, cite

We have included the information about the coverage and about the sequences of T. occidentalis, and corrected the mistake of 9GB, that was only 4,5GB and hence only 2x coverage. The genome coverage of 2 x was obtained from genome size of T. occidentalis published by Real et al. 2020, supposing both sister species have similar genome size. The sentence has been rephrased:

4.5 Gb of sequences were obtained from T. aquitania genome, corresponding to a coverage about 2 × of the genome, considering the genome of similar size to T. occidentalis genome (2.099Gb) [45], and 2Gb were obtained from T. occidentalis genome (approximately a coverage about 1 x)

  • Line 166: Was this genome assemble with long reads (CLR, HiFi, ultralong ONT)? This information is important, especially because you discuss the underrepresentation of satDNAs in the genome assembly

This information has been added in the new sentence:

Then, to check the presence of T. aquitania satDNAs on the closest relative species the consensus sequences were also masked on the chromosome-scale genome assembly of T. occidentalis, based on long- (PacBio) and short-read (Illumina) sequencing and scaffolded using Hi-C data, published by Real et al. [45] (GCA_014898055.1).

  • Line 172: Which statistical analyses? R is just a software, you need to detail this information

This information was detailed in Results section but we also included in Material and Method section as the reviewer suggested.

Now the sentence is:

Calculations and statistical analysis as correlation between variables using Sperman’s rank correlation rho and comparison between paired samples using Wilcoxon signed rank exact test, were performed in R base v.4.0.1 [52]

Results and Discussion

 Line 205: Something is wrong here. Multiplying 30M reads by 150bp, generate 4.5Gb.

The review have reason, we have one mistake in our data, we do not have 9 Gb of sequences we only have 4.5 Gb. The data have been corrected along the manuscript.

  • Line 218: What is "deeply analyzed"? Please, provide a more detailed information, especially because this is the most important step of the pipeline (i.e. determining the monomer sequences of satDNAs)

We have include a new sentence;

For each candidate cluster, we examined the contigs assembled by RepeatExplorer to search tandem repeated structures using the Dotmatcher and multiple-sequence align-ments and manual inspection to determine the consensus sequences.

  • Line 222: Names are different TaquSat or TaquiSat? Please, adjust this and check throughout the text

OK, checked and corrected in all the document

  • Line 223: correct “tow” by “two”

OK, done

  • Line 236: correct “thee”

OK, done

  • Lines 248-251: “we identified two different monomer length 437 and 466, both monomerS shared a fragment with an identity of 78.21%, and the remaining portion of the longer monomer corresponded with the first portion of a new the monomer repetition.” How this was done? On the contigs of RepeatExplorer? Which software? Manually? That's why authors should describe better the performed analyses to recover monomers of satDNAs

The determination of the two consensus sequences was performed as indicated in the new sentence includes in the results and discussion section

Also we indicated in the corresponding sentence that is changed to:  

However, in the satDNA TaquSat4-437-466, analyzing the contigs alignments and manually, we identified two different monomer length 437 and 466, …..

  • Line 263: I think we should interpret this specific result with care. Considering that monomer size of TaquSat5 is 3102bp-long, it is expected that TSI is the highest, mainly because fragments of Illumina sequencing are usually 300bp-long. So, the 0.99 value of TSI is most likely due to mapping in the same monomer, not mapping in the same satDNA adjacent monomer. Authors should consider this and highlight this on the text

Thanks to the reviewer to point out. It is completely true and the text has been modified accordingly.

We have included a new sentence:

However, in satDNA families with large monomer size, high values of TSI index could be related more to the size than the structure since short reads used in that calculation could correspond to the same monomers and not to tandem monomers.

  • Line 295: correct “populations”

OK, done

  • Line 301: Did you take into account the divergence values? When stating that TaquSat8 represents 0.69% of T. occidentalis genome, are all the divergence values included? If so, this should be indicated on the methods section. From what I remember, RepeatMasker outputs divergence values until 70%.  

Yes, we did. The reviewer is correct, the calcDivergenceFromAlign.pl outputs divergence values until 70%, and all those were considered for abundance calculations, although divergence values larger than 50% were 0.

  • Line 380: if all chromosomes are biarmed, NF would be 68, right?

The review has reason for NF (fundamental number) is 68, but we are indicating the NFa (autosomal fundamental number) that is used commonly, in that case, the sex chromosomes arms are not included, hence is 64.

Reviewer 2 Report

Presented article is dealing with characterization of repetitive and in particular satellite sequences in newly described species of Talpa genus, T. aquitania, native in broader Pyrenees region.

New generation sequencing on Illumina platform served as a pivotal step in this study. Obtained sequential data were processed by bioinformatic tools and satellite repetitive sequences were spotted and characterized. Presentation of their features in form of satellitome landscapes integrates their abundance with level of degeneration into one plot, giving this way possibility to judge to certain level evolutionary history of each of these satellite DNA families. Based on consensus sequence, probes for FISH in situ hybridization in T. aquitania karyotype were prepared. All these findings are compared to closely relative species Talpa occidentalis, as well as they are projected to the whole Talpidae family for much broader comparison. Accumulation of Sat families DNA into centromeric regions is one of the findings from the karyotyping combined with FISH.

  Overall, the research report is interesting, bringing new pieces of information not only concerning satellitome of Talpa aquitania, but also placing these new findings into broader context within Talpidae family. Also, general agenda concerning satellitome is addressed such as the issue of underrepresentation and bias of this category of DNA in genome assembly, when compared to sequencing data.

 However, some minor changes in manuscript should be done:

 234 - 235 Together with the next five families in the list and the telomeric sequences, they all comprise 1.13% of thee genome, while the remaining 11 satDNAs represent only 0.11%.

 322 It was interesting, however, that satDNA families with leptpkurtic distribution on their satellite landscape on T. aquitania, recently expanded families, were low abundant with larger degeneration value (TaquSat11-71), low homogenization index (TaquSat5-3102) or even not present (TaquSat8-45) on T. occidentalis

Author Response

Thank you for the revision of the manuscript, all your correction changes have been performed.